# Comparative Study of the Relationship between Microstructure and Mechanical Properties of Aluminum Alloy 5056 Fabricated by Additive Manufacturing and Rolling Techniques

**DOI:** 10.3390/ma16124327

**Published:** 2023-06-12

**Authors:** Alexey Evstifeev, Darya Volosevich, Ivan Smirnov, Bulat Yakupov, Artem Voropaev, Evgeniy Vitokhin, Olga Klimova-Korsmik

**Affiliations:** 1Mathematics and Mechanics Faculty, Saint Petersburg State University, Saint Petersburg 199034, Russia; ad.evstifeev@gmail.com (A.E.); i.v.smirnov@spbu.ru (I.S.); 2World-Class Research Center, State Marine Technical University, Saint Petersburg 119991, Russia; dasha.volosevich@mail.ru (D.V.); t-voropaev94@mail.ru (A.V.); evitokhin@yandex.ru (E.V.); o.klimova@ltc.ru (O.K.-K.)

**Keywords:** additive manufacturing (AM), wire arc additive manufacturing (WAAM), microstructure, 5000 series alloys, mechanical characteristics

## Abstract

In recent years, additive manufacturing of products made from 5000 series alloys has grown in popularity for marine and automotive applications. At the same time, little research has been aimed at determining the permissible load ranges and areas of application, especially in comparison with materials obtained by traditional methods. In this work, we compared the mechanical properties of aluminum alloy 5056 produced by wire-arc additive technology and rolling. Structural analysis of the material was carried out using EBSD and EDX. Tensile tests under quasi-static loading and impact toughness tests under impact loading were also carried out. SEM was used to examine the fracture surface of the materials during these tests. The mechanical properties of the materials under quasi-static loading conditions exhibit a striking similarity. Specifically, the yield stress $\sigma_{0.2}$ was measured at 128 MPa for the industrially manufactured AA5056_IM and 111 MPa for the AA5056_AM. In contrast, impact toughness tests showed that AA5056_AM KCV_full_ was 190 kJ/m^2^, half that of AA5056_IM KCV_full_, which was 395 kJ/m^2^.

## 1. Introduction

In recent years, Wire and Arc Additive Manufacturing (WAAM) has emerged as a groundbreaking and revolutionary technique in the world of advanced manufacturing processes. WAAM effectively combines the benefits of traditional casting technology while significantly reducing the necessity for machining monolithic wrought materials [1,2,3,4]. This innovative method has gained recognition for its cost-effective production of large-scale structures, particularly in the context of aluminum materials. The WAAM process is centered around the melting of a feed wire under the influence of an electric arc, resulting in the deposition of metal droplets in layers. The distinctive melting and solidification characteristics of WAAM play a vital role in determining the final material’s properties, including process-related defects such as porosity, cracking, and microstructure formation.

The research and development of various aluminum alloys using WAAM technology have accelerated in recent years, with several studies being conducted to explore their potential applications and benefits [5,6,7]. Among these alloys, the Al-Mg system–specifically the 5000 series aluminum alloys–has attracted significant interest from the scientific and industrial communities due to their remarkable strength and functional properties. The primary alloy element in the 5000 series aluminum alloys is magnesium, which enhances corrosion resistance in marine environments, making it a highly suitable candidate for applications in the shipbuilding industry and other sectors requiring corrosion-resistant materials. The extraordinary properties of 5000 series alloys in WAAM materials have been reported by numerous scientific groups, with studies focusing on 5056 [8,9], 5087 [10,11], 5183 [12,13,14,15], and 5356 [16,17,18] alloys.

The study [8] presents a theoretical model and experimental verification of the evaporation of alloying elements, such as magnesium, during wire-arc deposition of aluminum alloy 5056, revealing variations in chemical composition across the deposited layers. Authors of [9] developed a high-performance controlled short-circuiting metal transfer process for WAAM using a 5056 alloy.

One of the most critical concerns in additively manufactured aluminum alloys is porosity, which can negatively impact the material’s mechanical properties. Pores reduce the effective loading area and the material’s bearing capacity, serving as nucleation sites for crack initiation during fracture [19]. As a result, understanding porosity and its effects on mechanical properties has become a central focus in the research of additively fabricated aluminum alloys. A recent study [20] investigated the influence of WAAM + in-situ interlayer rolling hybrid technology on defect formation and mechanical properties of Al-4.5 Mg aluminum alloy. The findings showed that interlayer rolling effectively reduced porosity, resulting in defect counts being four times lower than in conventional WAAM production. Additionally, interlayer rolling contributed to a decrease in grain size from 59 μm to 23 μm, which could be caused by dynamic recrystallization [21], leading to a 30% increase in relative elongation while maintaining comparable strength properties. The fatigue limit of the material subjected to interlayer rolling was determined to be 86 MPa with a 95% survival rate. Numerical modeling revealed that “non-melting” defects had a more detrimental effect on final material properties due to their irregular shape and consequent higher stress concentration.

Despite the promising findings in the aforementioned studies, it is essential to recognize that the majority of experiments supporting the superior properties of these materials have been conducted under quasi-static loading conditions, with dynamic loading being largely overlooked. A comprehensive evaluation of the potential applications of AM materials necessitates the determination of their strength properties and an in-depth examination of their fracture behavior under various loading conditions, including dynamic loads.

The present study’s motivation is to broaden the understanding of aluminum alloys produced through WAAM under different loading scenarios. Specifically, 5056 aluminum wire was chosen as the raw material to assess strength characteristics under static and dynamic loading modes for both WAAM and rolling samples. The additive manufacturing material (AA5056_AM) is compared with the industrial material produced using the conventional method of rolling (AA5056_IM), which is a common mass-produced product that is readily available for purchase. The primary objective of this research is to elucidate the fundamental understanding of the influence of WAAM material’s microstructure features on dynamic strength, as well as to examine the effect of different manufacturing techniques on the mechanical behavior of aluminum alloys.

## 2. Materials and Methods

### 2.1. Raw Materials

Samples for studying structure were treated with SiC paper of P1200 and P2500 grit and then polished with diamond suspensions of 9 and 3 μm grit. The samples were then polished with a 0.25 μm SiO_2_-based colloidal suspension.

The studies were performed using commercial aluminum alloy 5056 produced by the typical method of rolling and manufactured by WAAM. The initial material for WAAM was supplied in the form of a wire with a diameter of 1.2 mm. Figure 1 shows the SEM image of the wire cross-section, as well as a map of the distribution of chemical elements within it. The wire has single pores that are 3 µm in size. The initial material contains spherical Mg_2_Si inclusions that are 1–5 µm in size, as well as intermetallic spherical Al6(Fe,Mn) inclusions that are 1–7 µm in size. 

The chemical composition of aluminum alloy 5056 under different conditions, as determined using EDX analysis (in %), is shown in Table 1.

The rolled sample contains a greater amount of intermetallic Al6(Fe,Mn), resulting in an increase in the content of Fe and Mn. The content of other elements is calculated from the condition of normalization of the total content of elements by 100%.

### 2.2. WAAM Process

The samples were manufactured using a WAAM installation produced by the Institute of Laser and Welding Technology (Saint Petersburg Marine Technical University, Saint Petersburg, Russia). The WAAM installation consists of a Kuka KR 60 HA 6-axis robot with an EWM 4X HP drive feeder. The welding source was an EWM alpha Q 551 Expert 2.0 puls MM FDW. The wire was fed to the work area via a TBi AUT-8W torch. The welding source, wire feeder, and robot arm are coordinated through an I/O module Busint X11. A schematic representation of the direct arc growth process is shown in Figure 2.

The presented technology of additive manufacturing is based on the principles of electric arc welding in a protective gas consumable electrode (MIG). The pulse mode of the source was used for the most stable transfer of the electrode material through the arc gap.

The processing parameters were obtained from a multifactorial experiment on the deposition of walls with a height of 20 layers. In the course of its implementation, the influence of current strength, voltage, and travel speed (TS) on the geometric parameters (width and height) and shaping of the deposited object was evaluated. A feature of multifactorial experiments is the repetition of the central point of the plan to eliminate errors. The technological parameters presented below were the center of the use plan, and therefore, have a high repeatability. When depositing six samples, the error in geometric dimensions is no more than 4%. Welding current, voltage, and travel speed were equal to 150 A, 22 V, and 20 mm/s, respectively. The pulse parameters were set automatically according to the internal program of the EWM source. The interpass temperature was controlled and the production of the next layer began once the previous bead had cooled to 50 °C. Dry compressed air was used to lower excessive temperatures.

### 2.3. Tension and Impact Toughness Tests

Tension tests were performed on a testing machine Shimadzu AG-50kNX at a strain rate of 0.001 1/s. Plane dog-bone samples with a gauge length of 5 mm and a width of 2 mm were used (Figure 3b). According to the tensile tests, average values of the yield stress ($\sigma_{0.2}$) corresponding to 0.2% of deformation, the ultimate tensile strength ($\sigma_{\mathrm{UTS}}$), the relative elongation to failure (δ), and the relative uniform elongation (δ_1_) were determined. 

Impact toughness was tested using the Charpy test using an Instron CEAST 9350 drop hammer according to standard GOST 9454 [22] on samples with a V notch. The samples with a length of 55 mm, height of 8 mm (working part 6 mm), and width of 5 mm (Figure 3a) were used. The speed upon impact was 5 m/s, corresponding to an impact energy of ~68 J. KCV*, KCV_fr_, and KCV_full_, representing the specific work before fracture initiation, the specific work after fracture initiation, and the total specific work, respectively, were determined.

The sample surface was polished to a slurry particle size not greater than 1 μm. To confirm the repeatability of the results, at least three samples were considered for each type of test.

### 2.4. Structural Investigation 

Fracture surfaces were examined using the scanning electron microscope (SEM) Zeiss AURIGA at an accelerating voltage of 10 kV. Electron backscatter diffraction (EBSD) patterns were obtained using Tescan Mira 3 scanning electron microscope with an Oxford Instrument EDX analysis attachment and an EBSD attachment at a voltage of 20 kV. The scanning step was 7 µm and the percentage of zero solutions was 2.46% for the deposited sample and 1.26% for the sample obtained by rolling. Samples for EBSD analysis were subjected to additional vibration polishing for 24 h using colloidal suspension.

## 3. Results and Discussion

### 3.1. Microstructure Parameters

The results of the EBSD analysis are illustrated in Figure 4, which provides a visual representation of the grain structure and orientation in both AA5056_AM and AA5056_IM materials. The images in Figure 4 reveal the distinct differences in grain size and morphology between the two materials, highlighting the impact of the manufacturing method on the microstructure of the aluminum alloy. Additionally, the clear presence of pores can be observed in AA5056_AM but not in AA5056_IM.

Table 2 complements Figure 4 by presenting quantitative data on the grain size and shape factors of AA5056_AM (bead boundary and bead body) and AA5056_IM materials. This information allows for a more detailed comparison of the grain structure between the two manufacturing methods. 

The grain size in the cross-section of the samples was calculated. The studied image was obtained at a magnification of 150×, and the size of the studied area was 2.4 × 2.4 mm. Only grains belonging entirely to the study area were included in the calculation. The length and width of a single grain were calculated using ImageJ software. To determine the degree of grain elongation, the shape coefficient, *h*, which is the ratio of grain length to width, was calculated. The closer *h* is to one, the more equiaxed the grain shape. The average values of grain size and shape coefficients were calculated from at least 150 measurements.

To further elucidate the grain size distribution in the two materials, Figure 5 displays histograms that showcase the frequency of occurrence of various grain sizes in the AA5056_AM and AA5056_IM samples. These histograms enable a comprehensive understanding of the variations in grain size within each material and provide a visual representation of the statistical distribution of grain dimensions.

The grain structure of the sample obtained using the WAAM technology has a periodic character. The grains at the bead boundary have a size of 114 μm and have a columnar character with a shape coefficient of 2.4. In general, the grains formed at the bead boundary are oriented toward the center of the cladding beads. As they move away from the bead boundary, the grains become finer, and in the bead body, they are 70 µm in size. In addition, the grains in the bead body are nearly equiaxial in shape, with a shape coefficient of 1.5. The formation of an uneven grain structure is typical of samples obtained by direct arc growth. High heat input, high-temperature gradients, and repeated cyclic heating are associated with the WAAM technique. The formation of coarse grains at the interface is promoted by the high heat input combined with repeated cyclic heating of the previous layer. The high-temperature gradient is responsible for a certain direction of grain growth, resulting in the formation of columnar grains at the boundary, oriented toward the center of the roll, which is typical for additive manufacturing [23,24]. The grain structure of the rolled product is represented by columnar grains that are 52 µm in size, elongated in the direction of processing. Compared with the grain structure of the WAAM samples, the grain shape coefficient was 3.3, indicating the formation of more elongated grains.

### 3.2. Tensile Test Results

From the in-depth analysis of the data presented in Table 3 and the stress-strain diagram shown in Figure 6, it can be observed that the AA5056 aluminum alloy material produced by additive printing utilizing the WAAM technology exhibits mechanical properties that are strikingly similar to those of the industrial alloy manufactured through conventional casting and rolling methods. 

Although the strength and ductility values are closely related, there are noticeable differences in the fracture surface of the samples subjected to uniaxial tension. The industrially manufactured material, AA5056, is characterized by a uniform distribution of pits on the fracture surface formed during the plastic flow of the metal (Figure 7a,b). In this case, particles of secondary dispersed phases Al6(Fe, Mn) [25] are responsible for the formation of microcavities. In contrast, there are significant differences in the fracture surface of the samples produced using the WAAM technique. Apart from micropores formed during the plastic flow on the concentrator as a secondary phase, there are larger pits (Figure 7c,d) that are a result of manufacturing defects, which can be identified in the UBSD map (Figure 4a).

It is important to consider these differences in fracture surface morphology when assessing the mechanical performance and reliability of additively manufactured components. The presence of manufacturing defects such as larger pits can potentially lead to premature failure or reduced fatigue life in the material, impacting its suitability for various applications.

### 3.3. Impact Toughness Test Results

The detailed analysis of the fracture behavior during impact toughness testing offers valuable insights into the materials’ performance under dynamic loading conditions. The use of force sensors on the drop weight impact testing machine ensures accurate measurements of load variations throughout the testing process, allowing for a better understanding of the materials’ response to these dynamic conditions. An in-depth examination of these measurements highlights the differences between AA5056_IM and AA5056_AM samples, providing a comprehensive comparison of their performance under similar test conditions. This information is crucial for understanding the mechanical behavior of additively manufactured materials and identifying areas for potential improvement.

The load and deformation work profiles presented in Figure 8 display the dynamic behavior of both AA5056_IM and AA5056_AM samples during impact toughness testing. The solid lines correspond to the load, while the dashed lines represent the deformation work (A) normalized by the cross-sectional area of the sample (S). The diagram shows an initial increase in load, reaching a maximum at time *t** when the stresses within the sample cause the material to fail. 

The impact toughness can be calculated using the formulas:(1)$$KCV^{*}=\frac{\int_{0}^{t^{*}}F\left(t\right)U\left(t\right)dt}{S}$$
(2)$$KCV_{full}=\frac{\int_{0}^{t_{full}}F\left(t\right)U\left(t\right)dt}{S}$$
(3)$$KCV_{fr}=KCV_{full}-KCV^{*}$$
where *F* is force, *U* is displacement, *S* is the cross-sectional area of the sample, and $t_{full}$ is the time of complete sample failure.

This fracture results from the development and propagation of micro and macro defects, which significantly reduce the specimen’s resistance to the applied load. Determining the time of fracture onset allows for the estimation of both pre-fracture work and fracture work, as illustrated in Table 4.

The data presented in Table 4 demonstrate that the AA5056_AM samples required only half the energy to fracture compared to the industrial aluminum alloy AA5056_IM samples. Simultaneously, the energy before fracture for the AA5056_IM samples was 2.5 times higher than that of the AA5056_AM samples, indicating the significantly reduced performance of the additively manufactured samples in comparison to their industrially produced counterparts. This observation is further supported by the fact that even when the AA5056_AM samples had already failed, the AA5056_IM samples maintained their stability, emphasizing the superior performance of the latter.

A closer examination of the fracture surfaces formed during the three-point bending tests of notched AA5056_IM and AA5056_AM specimens (Figure 9) reveals differences in their fracture behavior. The vertical dotted lines in Figure 9a,c represent the notch boundary. The fracture surface of the AA5056_IM sample (Figure 9a) is characterized by a shallow dimpled microrelief, which, upon magnification (Figure 9b), displays a uniform distribution of pits that are indicative of a ductile fracture. The bottom of these pits contains second-phase inclusions that led to their formation, contributing to the overall ductility of the material.

In stark contrast, the fracture surface of the AA5056_AM sample (Figure 9c) significantly differs from that of the AA5056_IM sample. Numerous pores, cracks, and large pits with an average diameter of 80 µm are visible on the fracture surface, occupying approximately 20% of the total area. These features collectively result in the poor impact toughness performance of the AA5056_AM samples due to the presence of interlayer defects. The surface of these pits appears smooth, devoid of visible defects, unlike those observed in the uniaxial quasi-static specimens.

This observation leads to the hypothesis that the micropores formed during the material deposition process may act as fracture concentrators under impact loading conditions. These defects, associated with high-stress concentrations, could potentially accelerate crack nucleation, leading to reduced impact toughness performance. Inclusions or voids can also introduce local variations in material properties such as density, stiffness, or toughness, resulting in an uneven stress distribution throughout the material. This non-uniformity in stress distribution could cause localized deformation, plastic flow, and ultimately, crack nucleation at a lower energy level than that observed for the rolled samples.

Thus, the findings of this study underline the importance of understanding the impact of additive manufacturing on the mechanical properties of materials, especially under dynamic loading conditions. Identifying and addressing the factors that contribute to the reduced impact toughness performance of additively manufactured materials is critical for their widespread adoption across various industries and applications. The results of this study demonstrate that the WAAM technique holds promise for the production of aluminum alloy components with mechanical properties comparable to those of conventionally manufactured materials. However, the presence of manufacturing defects, such as larger pits, underscores the need for a better understanding of the relationship between the additive manufacturing process, material microstructure, and resulting mechanical properties. Further research is required to optimize the WAAM process and minimize the occurrence of defects to ensure the production of high-quality, high-performance aluminum alloy components. Another aspect to be considered in future research is the investigation of the long-term performance of additively manufactured aluminum alloy components, including fatigue behavior and corrosion resistance. It is important to understand how the unique microstructures and defect characteristics of additively manufactured materials influence their long-term durability and resistance to environmental degradation. This information will be crucial for ensuring the reliability and safety of components produced using WAAM technology in various applications and operational environments.

## 4. Conclusions

In conclusion, this comprehensive study investigated the mechanical properties, microstructure, and fracture behavior of AA5056 aluminum alloy produced using Wire Arc Additive Manufacturing (WAAM) technology in comparison to the industrially manufactured AA5056 alloy via conventional casting and rolling methods. Through a series of experiments, the characteristics of both materials were thoroughly analyzed, with the aim of understanding the potential advantages and limitations of the WAAM technique for manufacturing high-performance aluminum alloy components.

The microstructure analysis revealed significant differences between the two AA5056 materials. The additively manufactured material displayed a periodic grain structure with columnar grains at the bead boundary and finer grains within the bead body. In contrast, the industrially manufactured material exhibited elongated columnar grains in the direction of processing. This difference in microstructure is attributed to the high heat input, temperature gradients, and repeated cyclic heating associated with the WAAM technique, which contributes to the formation of an uneven grain structure.

The quasi-static mechanical properties of the additively manufactured AA5056 material were found to be similar to those of the industrially manufactured alloy. Both materials exhibited comparable yield stress, ultimate tensile strength, and elongation values, indicating that the WAAM technique is capable of producing aluminum alloy components with quasi-static mechanical properties comparable to those of conventionally manufactured materials. 

Despite the close quasi-static properties of the materials, the study of the impact toughness behavior of the AA5056 materials revealed that the additively manufactured material required half the energy to fracture compared to the industrially manufactured alloy. This difference in impact toughness performance can be attributed to the presence of interlayer defects, such as pores and cracks, in the additively manufactured material. 

Future work should focus on developing strategies to mitigate the formation of defects and optimize the grain structure in the additively manufactured material. This could include refining the processing parameters, such as heat input and temperature gradients, as well as exploring post-processing techniques, such as heat treatments or hot isostatic pressing, to improve the material’s microstructure and mechanical properties. 

## Figures and Tables

**Figure 1 materials-16-04327-f001:**
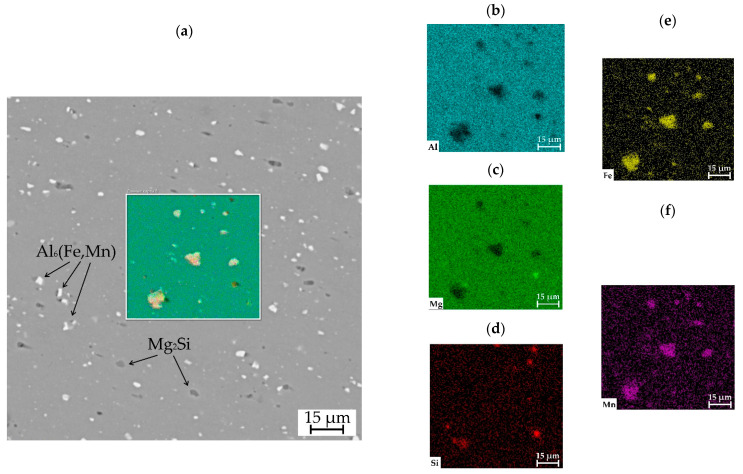
SEM-image of the wire cross-section: (**a**) general map; (**b**) Al map distribution; (**c**) Mg; (**d**) Si; (**e**) Fe; (**f**) Mn.

**Figure 2 materials-16-04327-f002:**
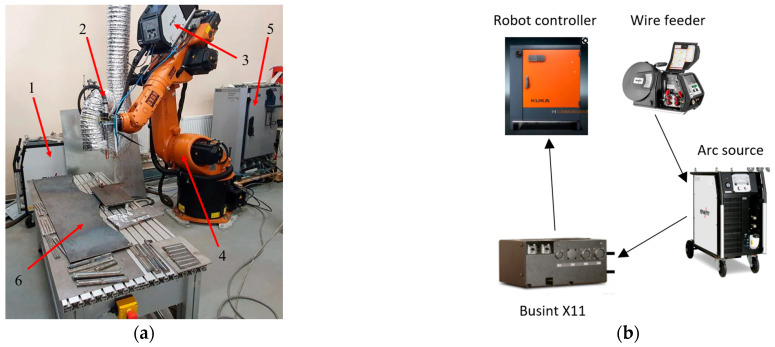
The WAAM station. (**a**) Appearance with components. 1: arc source; 2: arc torch; 3: wire feeder; 4: robot; 5: robot controller; 6: welding table; (**b**) Connection diagram.

**Figure 3 materials-16-04327-f003:**
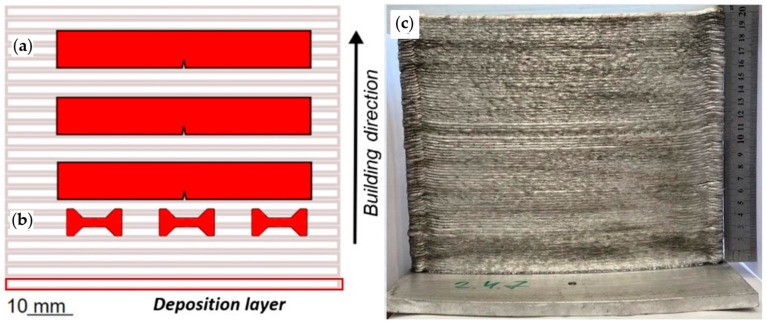
Schematic diagram of the shape and proportions of samples: (**a**) impact toughness; (**b**) tension; (**c**) photo of the manufactured sample.

**Figure 4 materials-16-04327-f004:**
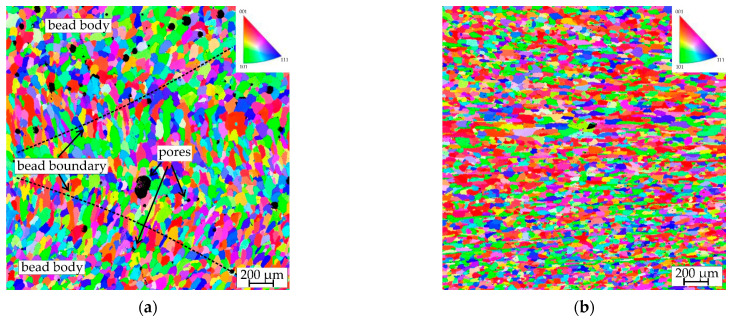
EBSD maps. (**a**) AA5056_AM; (**b**) AA5056_IM.

**Figure 5 materials-16-04327-f005:**
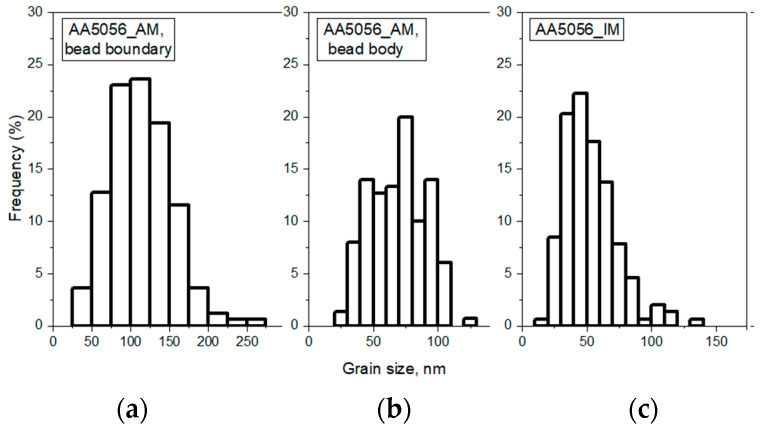
Grain size distribution (**a**) AA5056_AM, bead boundary; (**b**) AA5056_AM, bead body; (**c**) AA5056_IM.

**Figure 6 materials-16-04327-f006:**
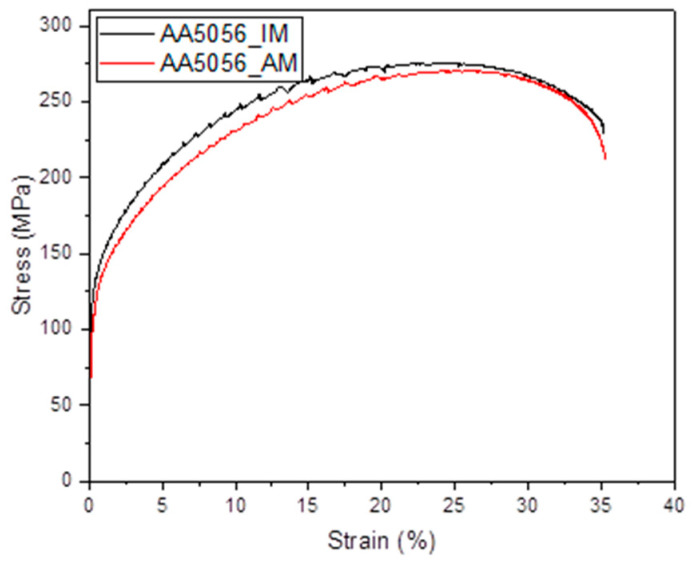
Stress-strain diagrams of AA5056_AM and AA5056_IM.

**Figure 7 materials-16-04327-f007:**
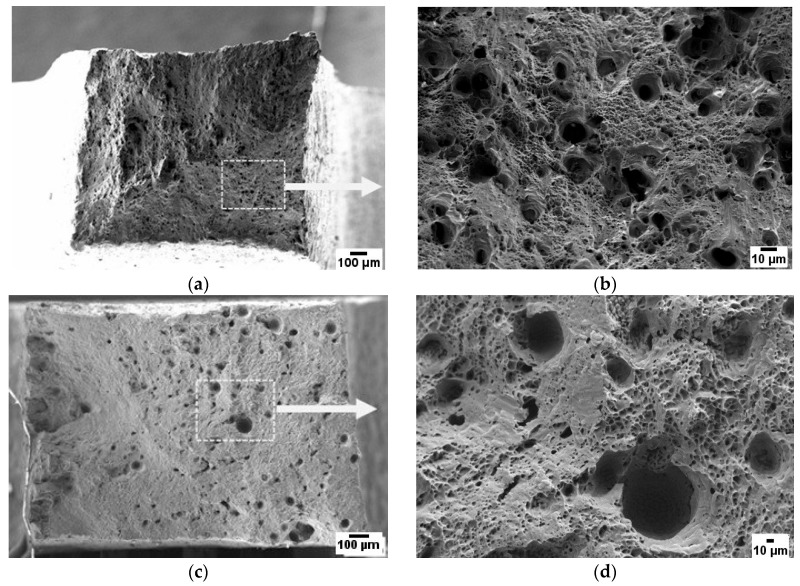
Fracture surface after tension fracture. (**a**,**b**) AA5056_IM; (**c**,**d**) AA5056_AM.

**Figure 8 materials-16-04327-f008:**
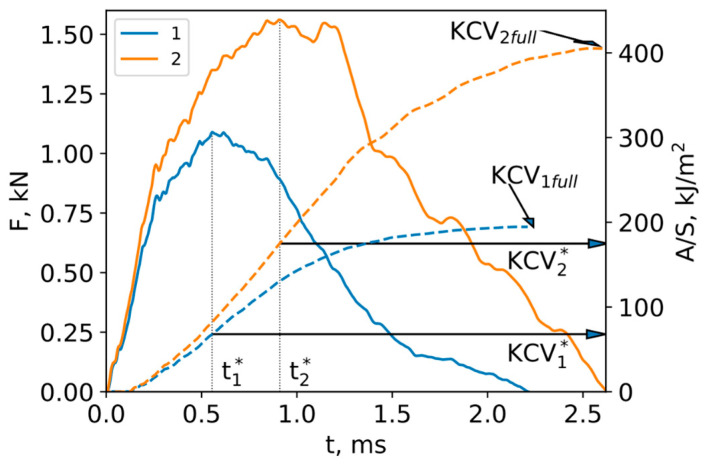
Diagram of impact toughness testing of (1) AA5056_IM and (2) AA5056_AM for samples with a V-type concentrator. Solid lines represent the loads; dashed lines represent deformation work A associated with the cross-sectional area of sample S.

**Figure 9 materials-16-04327-f009:**
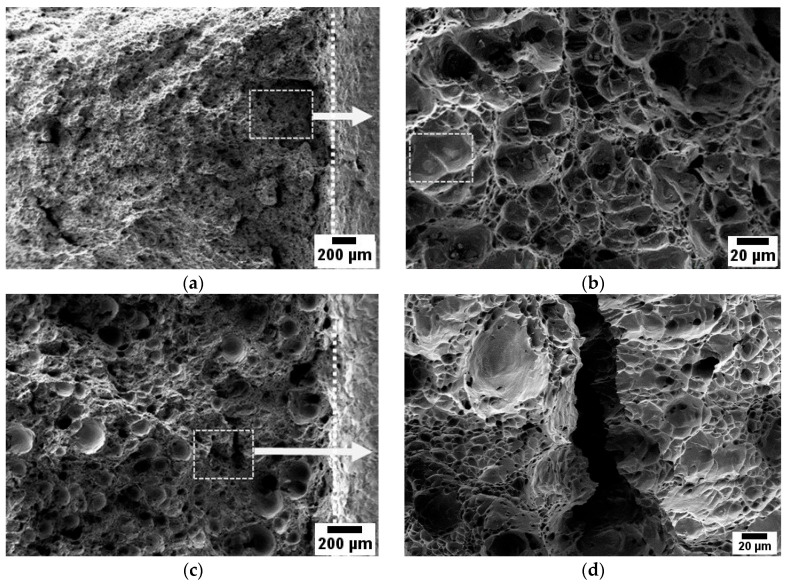
Fracture surface after impact toughness tests. (**a**,**b**) AA5056_IM; (**c**,**d**) AA5056_AM.

**Table 1 materials-16-04327-t001:** Chemical composition of materials under the initial condition and manufactured by the WAAM method according to EDX analysis (in Al balance, wt.%).

Material	Mg	Si	Ti	Mn	Fe	Cu	Zn
Wire	4.98	0.21	0.08	0.61	0.19	0.04	0.19
AA5056_AM	4.94	0.11	0.14	0.14	0.09	0.02	0.02
AA5056_IM	4.13	0.21	0.06	0.53	0.42	0.02	0.07

**Table 2 materials-16-04327-t002:** The results of grain size measurements of AA5056_AM and AA5056_IM.

Material	Grain Size, μm	Shape Factor h
AA5056_AM, bead boundary	114	2.4
AA5056_AM, bead body	70	1.5
AA5056_IM	52	3.3

**Table 3 materials-16-04327-t003:** Mechanical properties of materials under initial conditions and manufactured using the WAAM method. $\sigma_{0.2}\;$ is the yield stress, $\sigma_{UTS}$ is the ultimate tensile strength, δ is the relative elongation to failure, and $\delta_{1}$ is the relative uniform elongation.

Material	$\sigma_{0.2},\;\mathrm{MPa}$	$\sigma_{UTS},\;\mathrm{MPa}$	δ,%	$\delta_{1},\%$
AA5056_IM	128 ± 4	277 ± 1	33 ± 3	22 ± 1
AA5056_AM	111 ± 4	273 ± 4	33 ± 1	23 ± 1

**Table 4 materials-16-04327-t004:** Charpy impact toughness testing of AA5056_IM and AA5056_AM.

Material	*KCV**, kJ/m^2^	*KCV_Fr_*, kJ/m^2^	*KCV_Full_*, kJ/m^2^
AA5056_IM	166 ± 5	229 ± 1	395 ± 6
AA5056_AM	66 ± 1	122 ± 3	190 ± 3

## Data Availability

The data presented in this study are available on request from the corresponding author.

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
