# Peer review of "Comparative Study of the Relationship between Microstructure and Mechanical Properties of Aluminum Alloy 5056 Fabricated by Additive Manufacturing and Rolling Techniques"

_materials, 2023, doi:10.3390/ma16124327_

Round 1

Reviewer 1 Report

This comprehensive study investigates the mechanical properties, microstructure, and fracture behavior of AA5056 aluminum alloy produced using Wire Arc Additive Manufacturing (WAAM) technology compared to the industrially manufactured AA5056 alloy via conventional casting and rolling methods. The article is well-organized and written, and the images are high quality. Also, the article's title is practical and attractive, but the following points should be considered before publishing.

The abstract could be written better and needs minor revisions. The purpose of research and innovation should be clearly stated. Also, the performed tests should be presented first, and then the results should be presented quantitatively and qualitatively. The article needs general writing and grammar editing. Abbreviations could be used in the abstract.

The use of general sentences with more than four references can be seen in the introduction. For example, reference numbers 8-18 on line 42. On the other hand, appropriate references were not used to analyze the results. Although the introduction is long, it is written superficially in some paragraphs. Also, in the end, a suitable summary of the importance of the present issue should be provided.

Use the following resources to deepen the introduction. Microstructure evolution and deformation behavior during stretching of a compositionally inhomogeneous TWIP-TRIP cantor-like alloy by laser powder deposition. Effect of heat input on interfacial characterization of the butter joint of hot-rolling CP-Ti/Q235 bimetallic sheets by Laser + CMT. Gradient Structure of Ti-55531 with Nano-ultrafine Grains Fabricated by Simulation and Suction Casting.

How were the print parameters selected? Are these the optimal parameters? How was the reproducibility of the print capability checked? In the research method section, there is no mention of the selection of rolled and cast samples. In what conditions have these samples been compared with the welded sample? How many tensile and fracture test samples were prepared for each group? How are the reproducibility of mechanical properties results checked? How accurate was the strain measurement?

The results section is well organized and categorized. But some parts of it are just reporting the results. It is suggested to use the following sources. Hydrogen embrittlement behavior of SUS301L-MT stainless steel laser-arc hybrid welded joint localized zones. Probabilistic framework for fatigue life assessment of notched components under size effects. Effects of post-weld heat treatment on the microstructure and mechanical properties of laser-welded NiTi/304SS joint with Ni filler.

Table one should be added to the research method section, and a subsection titled raw material should be added. What is the reason for the difference in the chemical composition of the samples? The average grain size is calculated by what method? How to calculate toughness and energy absorption? Has the opening or crack growth been measured? Use the suggested sources to discuss the failure mechanism.

No comment.

Reviewer 2 Report

The article is a well-written, well-argued and intelligently outlined effort. As such there will be a few observations. Anyway, congratulations on your research 

1. I consider the title too long. I am convinced that the authors can find a more concise version of the title.

2. On line 178, I consider the phrase "The high temperature gradient is responsible for a certain direction of grain growth" a key statement, extremely important and partially unproven. I believe that the research of the article should have included the demonstration of this statement,

3. At line 197, I think that indeed in Fig. 6(a) a uniform distribution of pits can be observed, but in Fig. 6(c) I consider that the distribution of pits is gradual from left to right,

4. On line 222 I would add "...both AA 5506_IM (sample 1) and AA 5506_AM (sample 2)..."

5. On line 224: instead of "The diagrams show..." I would put "The diagram shows..."

6. In all figures 6 and 8, "mkm" is used as the unit of measurement of length. Please modify with "μm",

Reviewer 3 Report

In this manuscript, the authors compared the microstructure and mechanical properties of aluminum alloy 5056 fabricated by additive manufacturing and rolling. The following issues must be carefully addressed to publish in Materials:

1.      In the third paragraph of the introduction section, the authors reviewed some previous studies on the effect of in-situ interlayer rolling on structural and properties of WAAM samples, in which refined grain size was achieved after interlayer rolling. However, the relative refinement mechanism was not mentioned, which are suggested to be included in this part. The grain refinement was actually caused by dynamic recrystallization, which is similar to the microstructure evolution during thermal-mechanical processing like friction stir welding (https://doi.org/10.1007/s00170-022-09793-x, doi.org/10.1016/j.matdes.2022.111492), to which the authors can refer.

2.      Also in the introduction, the background explanation of why rolling samples were chosen for comparison with WAAM samples is not sufficient.

3.      The information of samples preparation procedure for microstructure investigation was missing.

4.      In Figure 4, the grain size and distribution were compared for two samples. However, another distinct difference in porosity between two samples was not analyzed.

5.      On page 7, line 202-204, it was claimed that ‘Apart from micropores formed during the plastic flow on the concentrator as a secondary phase, there are larger pits (Figure 6c, 6d) which are a result of manufacturing defects.’ What’s the manufacturing defects here? crack or pores?

6.      On page 8, the expression ‘the energy before fracture for the AA5056_AM samples was 2.5 times lower’ is not accurate and would cause confusion. It should be expressed as ‘the energy before fracture for the AA5056_IM samples was 2.5 times higher than that of the AA5056_AM samples’.

7.      The conclusion section is too long, which should be simplified.

English Writting needs polishing
